# Comparison between Retrograde Flexible Ureteroscopy and Percutaneous Nephrolithotomy for the Treatment of Renal Stones of 2–4 cm

**DOI:** 10.3390/medicina59010124

**Published:** 2023-01-08

**Authors:** Cozma Cosmin, Dragos Adrian Georgescu, Petrisor Geavlete, Razvan-Ionut Popescu, Bogdan Geavlete

**Affiliations:** 1Department of Urology, Carol Davila University of Medicine and Pharmacy, 050474 Bucharest, Romania; 2“Sfantul Ioan” Emergency Clinical Hospital, 042122 Bucharest, Romania

**Keywords:** flexible ureteroscopy, percutaneous nephrolithitomy, renal lithiasis

## Abstract

*Background and objectives:* Renal stones are widespread, with a lifetime prevalence of 10% in adults. Flexible ureteroscopy enables urologists to treat lower calyx stones or even complex renal stones through the natural orifice and achieve an acceptable stone-free rate. Hence, we analyzed the effectiveness and safety of FURS versus PCNL in treating renal stones between 20 and 40 mm in diameter. *Materials and methods:* We retrospectively analyzed 250 consecutive patients with large renal solitary stones (stone burden between 2 and 4 cm) from 1 January 2019 to 31 December 2020. The patients were divided into two groups: group 1 (125 patients), in which the patients were treated by a retrograde flexible ureteroscopic approach, and group 2 (125 patients), in which we used percutaneous nephrolithotomy. Stone characteristics and anatomical data were observed based on the computed tomography (CT) and/or KUB (Kidney-ureter-Bladder) radiography imaging archive. *Results:* The mean stone burden was 26.38 ± 4.453 mm in group 1 and 29.44 ± 4.817 mm in group 2. The stone-free rate after the first ureteroscopy was higher for the PNL(percutaneous nephrolithotomy) group (90.4%) than the F-URS group (68%). After two sessions of ureteroscopy, the SFR was 88.8% in the first group, and after three procedures, the SFR rose to 95.2%. The overall complication rate was higher in group 1 than in group 2 (18.4% vs. 16.8%), but without statistical relevance (*p* > 0.5). Furthermore, we encountered more grade III and IV complications in the PNL group (8.8% vs. 4.8%, *p* < 0.05). *Conclusion:* Flexible ureteroscopy proves to be efficient in treating renal stones over 2 cm. However, the patients must be informed that more than one procedure might be necessary to overcome the entire stone burden.

## 1. Introduction

Renal stones are widespread, with a lifetime prevalence of 10% in adults. The global incidence is increasing due to obesity and diabetes. [1]. The most frequent reason for emergency admission to urology departments is flank pain from stones (renal colic). In approximately 35% of instances, stones form in the lower pole of the kidney, where they are least likely to pass naturally. Extracorporeal shockwave lithotripsy (ESWL), percutaneous nephrolithotomy (PCNL), and flexible ureterorenoscopy (FURS) with laser lithotripsy are the three current methods for removing lower pole kidney stones [2].

At present, percutaneous nephrolithotomy (PCNL) is recommended as the first-line treatment of choice for renal stones more than 2 cm in diameter [3]. Even if PCNL has high efficiency, it continues to have non-negligible morbidity effects, such as bleeding requiring angioembolization, urinoma, and organ injury, although rare [4,5]. FURS enables urologists to treat lower calyx stones or even complex renal stones through the natural orifice, and helps to achieve an acceptable stone-free rate because of technological advancements with laser lithotripsy systems and innovative endoscopic baskets [6]. In several investigations, the management of >2 cm renal stones was compared between PCNL and FURS. Concerns have been raised over whether FURS offers a stone-free rate (SFR) equivalent to that reached by PCNL, while posing fewer risks because of uneven results across individual controlled studies [7]. Hence, we analyzed the effectiveness and safety of FURS versus PCNL in treating renal stones over 20 mm in diameter.

## 2. Materials and Methods

We retrospectively analyzed 250 consecutive patients with large renal solitary stones (stone burden between 2 and 4 cm) from 1 January 2019 to 31 December 2020. The patients were divided into two groups: group 1 (125 patients), in which the patients were treated by a retrograde flexible ureteroscopic approach, and group 2 (125 patients), in which patients were treated by percutaneous nephrolithotomy. Stone characteristics and anatomical data were observed based on the computed tomography (CT) and/or KUBradiography imaging archive. For larger stones, more than one session of flexible ureteroscopy was necessary. The preoperative CT images calculated the stone surface area based on the largest volume (length × width  =  mm^2^). The follow-up evaluation was undertaken one month after the operation using plain film of KUB(Kidney-ureter-Bladder) for opaque stones, ultrasonography for non-opaque stones, or non-contrast CT of the whole abdomen (Figure 1A,B).

Success was defined as the absence of residual stone fragments larger than 3 mm at the end of the procedure.

The experimental intervention FURS will be compared with the standard intervention PCNL. Both interventions are currently in general use by urology departments throughout the world. The surgical interventions FURS and PCNL were carried out by a trained urologist or a trainee urologist under the supervision of a senior urologist.

Perioperative and postoperative complications, stone-free rate, operation time, and length of hospital stay were compared between the PNL and RIRS(retrograde intrarenal surgery) groups.

### 2.1. Surgical Procedures

For FURS, a small (3 mm diameter) flexible endoscope called a ureteroscope is inserted into the kidney through the urethra, bladder, and ureter to view the stone up close. For our procedure, an Olympus URF-V2 flexible ureteroscope(Olympus Medical System, Tokyo, Japan) was employed and a Dornier Medilas H Solvo 35 Holmium laser(Dornier MedTech, Munich, Germany) served as the power source. The kidney stone is subsequently broken up using laser energy after being fed a laser fiber through the ureteroscope’s operating channel (usually a 200 or 273 m holmium laser fiber)(Figure 2). Smaller fragments (2 mm) can be allowed to pass naturally, but larger fragments can be recovered using a wire basket device that is passed through the working channel. Typically, during the week following the treatment, the patient will pass any leftover fragments in the urine. The surgery normally necessitates an overnight stay and general anesthesia. At the conclusion of the treatment, a temporary ureteral double-J stent may be inserted to guard against ureter blockage brought on by swelling of the lining cells. The patient is usually discharged the day after the intervention, if they are clinically stable.

PCNL is a surgical procedure to remove stones from the kidney by a direct approach. A 10 mm incision is first created in the skin overlaying the kidney, and a needle is then inserted, guided by a fluoroscope, into the kidney’s collecting system. After that, contrast fluid can be delivered into the collecting system through a ureteral stent to direct the needle’s path through the skin and into the kidney. The needle is positioned to provide optimum access for successful removal of the stone using the imaging that is available, which is often a computed tomography scan of the kidneys, ureters, and bladder (CT KUB). For a stone in the lower pole of the kidney, placement is typically into the lower section of the collecting system. Once the needle has been successfully inserted, a flexible guidewire is inserted into the kidney’s collecting system and is used to direct the stretching (dilatation) of the needle track so that it is wide enough for a hollow rigid access sheath to be passed; this creates a 10 mm wide channel between the skin and the kidney’s urine collecting system. The stone can then be seen using a hard metal telescope (nephroscope), which is placed into the kidney’s collecting system. The stone can then be removed whole using graspers, or fragmented using a variety of energy delivery tools, most frequently an ultrasonic probe or pneumatic instrument. A tube put through the access channel or as a stent down the ureter into the bladder is used to temporarily empty the kidney after the procedure. In our situation, a nephrostomy tube was employed. A urinary catheter could also be implanted to temporarily drain the bladder after the treatment. Depending on the complexity, the procedure typically takes 1–3 h to complete under general anesthesia, and patients typically stay in the hospital for a few days. After 24–48 h, the drainage tubes are routinely removed without additional anesthetic. Prior to being released from the hospital, stone removal is checked both during the procedure and, if necessary, by a plain x-ray (or other imaging). The patient was discharged after the nephrostomy tube was removed and they were clinically stable, without hematuria or the presence of lumbar fistulae.

### 2.2. Data Analysis

All collected data were obtained from the medical records and further processed and illustrated using Microsoft Excel and Word available on Microsoft Office 18.2008.12711.0(Microsoft corp., Washington, DC, USA) and IBM SPSS Statistics Version 26(IBM, Chicago, IL, USA).

## 3. Results

Patient characteristics revealed that the mean age of subjects was 54.96 ± 14.283 (std. dev.) in group 1, and 57.04 ± 8.511 in group 2. The mean stone burden was 26.38 ± 4.453 mm in group 1, and 29.44 ± 4.817 mm in group 2. The stone location was mainly in the lower calyx in both groups, followed by the renal pelvis. The average operative time was 49.07 ± 4.558 min in the F-URS group, and 70.64 ± 7.306 in the PNL group, whereas the fluoroscopy time was higher in the PNL group (55.25 ± 11.987 s vs. 30.10 ± 7.914 s). The patients’ perioperative characteristics are displayed in Table 1.

The hospital stay was higher in the PNL group (162.62 ± 31.802 h) than in the F-URS group (78.14 ± 23.393 h).

The stone-free rate after the first ureteroscopy was higher for the PNL group (90.4%) than for the F-URS group (68%). After two sessions of ureteroscopy, the SFR was 88.8% in the first group, and after three procedures, the SFR rose to 95.2% (Table 2).

Complication rates were evaluated using the Clavien–Dindo modified system [8]. The overall complication rate was higher in group 1 than in group 2 (18.4% vs. 16.8%), but without statistical relevance (*p* > 0.5). Furthermore, we encountered more grade III and IV complications in the PNL group (8.8% vs. 4.8%, *p* < 0.05). No grade V complication was encountered in either of the groups. The complications encountered in both groups are listed in Table 3.

## 4. Discussion

### 4.1. Complementary Treatment

Four studies with data available for the combination reported the number of patients needing complementary treatment. Zewu et al.’s meta-analysis showed that the FURS group was associated with significantly more complementary therapies.

In Jiahua Pan et al.’s study, the mean procedure number was 1.18 in the RIRS group and 1.03 in the mPCNL group, respectively [9]. Furthermore, Yu Zhang et al. found that PCNL achieved 85.7% (36 of 42) on a one-session stone-free rate, whereas the RIRS group achieved 58.8% (20 of 34, *p* = 0.008) [10]. In addition, Pieras et al. determined that a larger number of auxiliary processes were needed for the ureteroscopy group (20%), than for the nephrolithotomy group (7%) (*p* = *0*.04) [11].

In Abdullah Erdogan’s study, the rate of patients requiring additional surgical intervention after the first operation was higher in the RIRS group, but this was not statistically significant (21.3%  >  9.1%; *p*  =  0.061) [12].

Concerning our study, although in group 1 there were patients who required more than one session of flexible ureteroscopy, the PNL group also required complementary treatment such as flexible ureteroscopy, double J stenting, and angioembolization. Excluding the multiple sessions of ureteroscopy in group 1, the overall complementary treatment rate was higher in the PNL group.

### 4.2. Stone-Free Rate

According to Zhu Zewu’s meta-analysis, the synthesis results showed that PCNL offered an initial SFR superior to retrograde FURS for managing 2–3 cm renal stones, consistent with most of the included studies. In addition to the inherent flaws of current FURS techniques and systems, such as constrained working channels and the flexibility of ureteroscopy, residual fragments are more likely to represent a cluster of clinically insignificant fragments. This is one explanation for one-stage FURS with a relatively lower SFR. Bryniarski et al. first described the technique by shifting the patient’s position to move lower pole stones, which improved the initial SFR of FURS [13]. Because dust can make it difficult to see the clean operative field and because it can be challenging to distinguish a small, fractured stone in the middle of dust, Mulţescu et al. and Cho et al. both suggested that dusting followed by fragmentation may be preferable for stones larger than 1 cm [14]. According to Kuo et al., short fibers (200–270 m) are preferable to larger ones (365 m) because they allow for fluid irrigation and flexibility without reducing fragmentation efficacy [15]. Chen et al. developed a novel technique to aspirate the pieces directly using vacuum aspiration UAS and artificial water circulation, made possible by infusing saline into the ureteral catheter’s tail end. This technique also helped save surgical time because it used fewer baskets [16]. We should be able to greatly increase the SFR of one-session FURS in the near future, thanks to advancements in laser fibers with higher energy transmission and the combination of improved flexibility and smaller diameter endoscopes.

Among the included investigations, there was discrepancy regarding whether FURS gave a final SFR equal to that attained by PCNL. One month after surgery, Zengin et al. reported that the PCNL group’s final SFR was 95.5%, while Chen et al. stated that the FURS group’s final SFR was identical to that of the PCNL group (89.1 vs. 92.5%). We can accept the discrepancy with caution because multisession FURS can result in a satisfied SFR, as shown by the most recent study, which found that single-session FURS had a satisfied SFR of 67.2%, and that the final SFR following multisession procedures was 89.1% [16]. This conclusion has been supported by similar investigations [13,17,18]. The surgeon’s judgment and each patient’s preferences also played significant roles in the choice of subsequent therapy. When treating renal stones larger than 2 cm in diameter, Kang et al. performed a meta-analysis comparing the ultimate SFR between FURS and PCNL. They discovered that PCNL was superior to FURS, with significant heterogeneity [19]. The various stone loads in the included studies could be the source of the inconsistency. Because the stone volume grows with an increase in diameter, Kang et al.’s earlier meta-analysis revealed a significant range in stone diameter, ranging from 2 cm to 4 cm or even greater.

In our study, the stone-free rate after the first ureteroscopy was higher for the PNL group (90.4%) than the F-URS group (68%). After two sessions of ureteroscopy, the SFR was 88.8% in the first group, and after three procedures, the SFR rose to 95.2%. Our results are consistent with the data found in the literature and confirm that it can be achieved at a decent stone-free rate after multiple sessions of flexible ureteroscopy.

### 4.3. Complications

Postoperative complications varied by group, but those specific to PCNL included pelvic perforation, persistent urine leakage, and bleeding that required transfusion or even embolization. On the other hand, FURS-specific ureteral harm and strictures existed. As a result, when we compared the Clavien-graded complications, we discovered that PCNL was linked to a much greater transfusion rate, which in turn led to a higher frequency of major complications and total complications. Particularly in individuals with solitary kidneys, the possible risk of bleeding resulting from PCNL should be seriously taken into account. When receiving PCNL, these patients have a higher chance of developing acute renal failure than those with bilateral kidneys because compensatory hypertrophy makes them more susceptible to bleeding which necessitates embolization [4].

Additionally, functional parenchymal loss following PCNL and urinary blockage by blood clots both impair the function of a single kidney [20]. In patients with solitary kidneys, Bai et al. observed that FURS was a safer alternative to PCNL after discovering that 11.7% (7/60) of those receiving PCNL experienced bleeding that required transfusion. The FURS group, however, had no transfusion-requiring patients (0/56) [21]. Giusti et al. and Shi et al. also supported the conclusion that FURS was sufficiently safe for use in individuals with solitary kidneys [20,22]. While they were directly associated to urosepsis with FURS, the significant consequences (Clavien grades III and IV) were closely related to severe bleeding attributable to PCNL. According to the data from the literature, there were no appreciable variations in major problems. The risk of urosepsis secondary to FURS was theoretically greatly increased by larger renal stones because they required significantly more time to remove. Sepsis would be hazardous and potentially fatal without quick therapy. According to Blackmur et al., postoperative urosepsis was strongly linked with positive urine culture and large stone size in a matched-pair study [23]. Therefore, it is crucial to have a preoperative microbiological test and keep the surgery time for the FURS operation between 90 and 120 min. Furthermore, there is evidence that high irrigation pressure is a risk factor for urosepsis in RIRS [24]. Unfortunately, we did not have the means to measure the intrarenal pressure, but we tried to reduce it by using a ureteral access sheath [25].

In our study, the overall complication rate was higher in group 1 than in group 2 (18.4% vs. 16.8%), but without statistical significance (*p* > 0.5). Furthermore, we encountered more grade III and IV complications in the PNL group (8.8% vs. 4.8%, *p* < 0.05). No grade V complication was encountered in either of the groups.

### 4.4. Operative Time

The operative time is an essential factor in the influence of postoperative complications. A longer operative time increases the risk of urosepsis secondary to FURS, and the need for blood transfusion secondary to hemoglobin decline in the PNL group. In the literature, some studies found opposing results to other studies. According to Murat Bagcioglu et al., the mean operation time of the RIRS group was significantly shorter than of the microperc group (55.62 ± 19.62 min and 98.50 ± 29.64 min, respectively, *p* < 0.001) [26]. In addition, Pieras et al. found that the surgical time was longer for the nephrolithotomy group (121 ± 52 min) than for the ureteroscopy group (93 ± 42 min) (*p* = *0*.004) [20]. According to He-Qun Chen et al., the F-URS group was similar to the mPNL group in terms of the mean duration of surgery (92.8 ± 26.1 vs. 87.4 ± 31.5 min, *p* = 0.137) [16].

As opposed to that, Kursad Zengin found that the mean operative time was 63 ± 22 min in the PNL group, and 81 ± 41 min in the RIRS group (*p* < 0.001) [18].

In our study, the average operative time was 49.07 ± 4.558 min in the F-URS group and 70.64 ± 7.306 in the PNL group, whereas the fluoroscopy time was higher in the PNL group (55.25 ± 11.987 s vs. 30.10 ± 7.914 s).

### 4.5. Hospital Stays

Concerning the hospital stay, Abdullah Erdogan found that the length of hospital stay was longer in the PNL group than in patients who underwent RIRS, which is similar to the data obtained by Akman et al. [21,27]. This can be explained by the presence of nephrostomy and the follow-up requirement after blood transfusion in PNL. Based on these results, we can state that for patients who undergo RIRS, less time is spent in the operating room and there is a shorter duration of patient bed occupation. Many studies indicated that PCNL was associated with a significantly longer hospital stay [16,18,20].

Our study presented the same results, finding that the hospital stay was higher in the PNL group (162.62 ± 31.802 h) than in the F-URS group (78.14 ± 23.393 h). In the PCNL group, the patient was discharged after the nephrostomy tube was removed and when they were clinically stable, without hematuria or the presence of lumbar fistulae, whereas in the FURS group, the patient was usually discharged the day after the intervention if they were clinically stable.

### 4.6. Cost Analysis

The cost analysis of flexible ureteroscopy and percutaneous nephrolithotomy is shown in Table 4 and Table 5. The operating fee and surgical equipment are higher for the flexible ureteroscopy, mainly because of the flexible endoscope, which is available for a limited number of procedures, in contrast to the nephroscope and the laser fiber. In our case, we used a single flexible ureteroscope for all 125 interventions, and utilized a laser fiber every 10 interventions. As expected, the postoperative and hospital stay costs were higher for PCNL. As a result, the cost for a FURS intervention was 1219 Euros, whereas the cost for PCNL intervention was 1789. Taking into consideration that in our study, 1.432 procedures were necessary to reach a stone free rate of 95.2% for FURS, it results that the cost to reach a stone free rate of 95.2% is 1745.608 euros in case of FURS, and the cost to reach a stone free rate of 90.4% is 1789 euros in case of PCNL.

### 4.7. Study Limitations

This study must be interpreted within its limitations. The retrospective nature of this study might precipitate potential bias.

## 5. Conclusions

In conclusion, flexible ureteroscopy proves to be efficient in treating renal stones over 2 cm. However, the patients must be informed that more than one procedure might be necessary to overcome the entire stone burden. Therefore, it is advisable to weigh up the risks and benefits in light of each patient’s unique characteristics, and to reach a decision together with the patient after outlining the pros and drawbacks of each surgery.

## Figures and Tables

**Figure 1 medicina-59-00124-f001:**
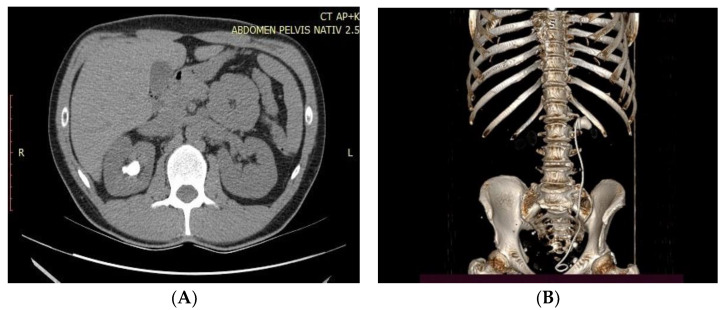
(**A**,**B**). CT scan image and reconstruction of large pelvic stone.

**Figure 2 medicina-59-00124-f002:**
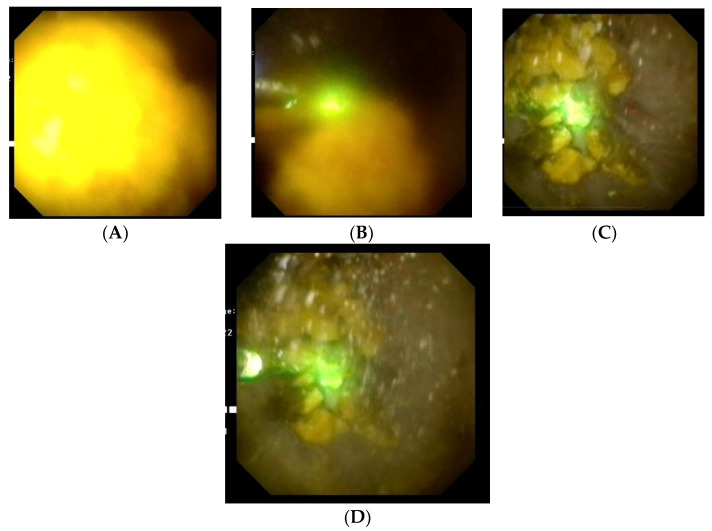
(**A**–**D**). Flexible ureteroscopy and Holmium laser lithotripsy of a large renal stone.

**Table 1 medicina-59-00124-t001:** Patients’ perioperative characteristics.

		F-URS Group	PCNL Group	*p*
Age		54.96 ± 14.283	57.04 ± 8.511	0.046
Mean Stone burden (mm)		26.38 ± 4.453	29.44 ± 4.817	0.0059
Stone location	Pelvis (number-nr.)	26	40	
	Upper Calyx (nr.)	24	6	
	Middle Calyx (nr.)	10	8	
	Lower Calyx (nr.)	75	71	
Operative time (min)		49.07 ± 4.558	70.64 ± 7.306	0.011
Fluoroscopy time (s)		30.10 ± 7914	55.25 ± 11.987	0.0047
Hospital stay (hours)		78.14 ± 23.393	162.62 ± 31.802	0.00231

**Table 2 medicina-59-00124-t002:** Stone-Free rate of F-URS and PCNL.

	F-URS Group			PCNL-Group
	1st session	2nd session	3rd session	
Stone-Free Rate	68%	88.8%	95.2%	90.4%

**Table 3 medicina-59-00124-t003:** F-URS and PCNL Complications.

	F-URS Complications	Nr. Cases	PCNL Complications	Nr. Cases	*p*
Clavien I	Fever	10	Fever	4	
	Mild Hematuria	16	Hematuria	8	
	Total	26	Total	12	0.012
Clavien II	Urinary tract infections	8	Significant bleeding requiring blood transfusion	6	0.035
Clavien III			Persistent urine leakage requiring double J ureteral stenting	4	
			Arterio-venous fistula requiring angioembolization	5	
			Total	9	0.022
Clavien IV	Sepsis requiring ICU management	6	Sepsis	2	0.0012
Total		18.4%		16.8%	>0.5

**Table 4 medicina-59-00124-t004:** Cost-analysis of FURS and PCNL.

Cost Analysis (Euro)	FURS	PCNL
Operating fee	300	200
Surgical equipment	244	189
Total procedural costs	544	389
Postoperative costs	25	50
Hospital stay	650	1350
Total	1219	1789

**Table 5 medicina-59-00124-t005:** Basic equipment cost for FURS and PCNL.

Basic Equipment Cost (Euro)	FURS	PCNL
Sterile drapes	40	60
Irrigation set	2	2
Syringes	1	1
Foley catheter	2	2
Sodium chloride 0.9% 2000 mL	4	4
Hydrophilic guidewire	15	30
Ureteral catheter	-	10
Nephrostomy tube	-	5
Laser fiber (one every 10 procedures)	100	-
Nephrostomy tract dilators	-	5
Nephrostomy sheath		10
Double-J catheter	20	-
Basket	60	60
Total	244	189

## Data Availability

Data supporting reported results can be found in the archived datasets of the Saint John Emergency Clinical Hospital, Bucharest.

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
