# Peer review of "Comparison between Retrograde Flexible Ureteroscopy and Percutaneous Nephrolithotomy for the Treatment of Renal Stones of 2–4 cm"

_medicina, 2023, doi:10.3390/medicina59010124_

Round 1
Reviewer 1 Report
The title should clearly written the range of the stone burden was 2 - 4 cm.
Factors that causes of sepsis in fURS includes intrarenal pressure caused by URS. It should be discussed and the authors could state their opinion(s) concerning this issue.
Overall this series is useful for the urinary tract stones reference, although only from single center.
Author Response
- Thank you for your comment. We have updated the title of the study "Comparison between retrograde flexible ureteroscopy and percutaneous nephrolithotomy for the treatment of renal stones of 2-4 cm".
- Thank you for your comment. Unfortunately we did not have the possibility to measure intrarenal pressure, but we added a paragraph in the "Discussion" chapter addressing to this concern.
Reviewer 2 Report
The authors compared the outcomes of retrograde flexible ureteroscopy and percutaneous nephrolithotomy for the treatment of renal stones over 2 cm. These results are to be expected. I have several comments and suggestions.
1. Please provide a table or two tables presenting the clinical characteristics and prioperative outcomes of the two groups.
2. Do you included patients with multiple stones? If a patient has several stones, how to calculate the stone burden in your study?
3. Please provide the costs analysis of the two groups.
4. The table 1 and 2 can be merged into one table, so that the comparison will be more intuitive.
5. The figures are a little messy and can be rearranged and edited.
Author Response
Dear Reviewer 2,
Thank you very much for your valuable comments to improve our manuscript.
1.Thank you for your comment. We updated the manuscript with two tables presenting the clinical characteristics and perioperative autcomes.
2. Thank you for your comment. We updated the manuscript stating that we included patients with solitary stone.
3. Thank you for your comment. Unfortunately we did not calculate the cost analysis of the two groups.
4. Thank you for your comment. We merged the two tables for a more intuitive comparison.
5. Thank you for your comment. We rearranged a little the pictures. Please tell us if we should change the pictures.
Round 2
Reviewer 2 Report
I would like to thank the authors for their efforts in revising the manuscript. But the problems 3 and 5 have not been well addressed.
1. I think the cost-analysis is a very important outcome in comparing PCNL and (several stages of) FURS.
2. The hospital stays are different between the two groups, the criteria of discharge should be provided in the methods.
3. Where are figure 1a and 1b? Figure 3-6 can be merged to one figure and labeled as a, b, c and d.
4. The parameters in table 1 and 3 can be statistically compared between the two groups and p values should be provided.
5.There are many "," in the numbers, they should be changed to "." .
Author Response
Dear Reviewer,
Thank you for the comments on improving our manuscript. We hope that we responded efficiently to the comments.
- Thank you for the comment. We added a subchapter in the "Discussion" chapter named "Cost analysis," where we calculated the cost for both the FURS and PCNL.
- Thank you for the comment. We defined the discharge criteria for the FURS and PCNL in the "Material and method" and "Discussion" chapters.
- Thank you for the comment. We renamed the figures as recommended. We also corrected the figures' names in the text.
- Thank you for the comment. We compared the parameters in tables 1 and 3 and added the p-value.
- Thank you for the comment. We corrected the "'," in all the numbers.